# The Effect of Meditation, Mindfulness, and Yoga in Patients with Rheumatoid Arthritis

**DOI:** 10.3390/jpm12111905

**Published:** 2022-11-15

**Authors:** Laura Slagter, Koen Demyttenaere, Patrick Verschueren, Diederik De Cock

**Affiliations:** 1Skeletal Biology and Engineering Research Centre, KU Leuven, 3001 Leuven, Belgium; 2Department of Neurosciences, Psychiatry Research Group and University Psychiatric Center KU Leuven, Belgium University Psychiatric Center, KU Leuven, 3001 Leuven, Belgium; 3Department of Rheumatology, University Hospitals of Leuven, 3001 Leuven, Belgium; 4Biostatistics and Medical Informatics Research Group, Department of Public Health, Faculty of Medicine and Pharmacy, Vrije Universiteit Brussel (VUB), 1005 Brussels, Belgium

**Keywords:** mind–body therapies, meditation, yoga, mindfulness, rheumatoid arthritis

## Abstract

Objectives: Mind–body therapies (MBTs), including meditation, yoga, and mindfulness, create an interaction between the mind and body to enhance health. MBTs are perceived by both patients and healthcare professionals as valuable in the management of rheumatoid arthritis (RA), but the extent of this contribution is unclear, as are the patient subgroups who benefit most from MBTs. Therefore, this systematic literature review investigates the effects of meditation, mindfulness, and yoga in patients with RA. Methods: We searched four databases (PubMed, Embase, Web of Science (core collection, Chinese and Korean collection), and CINAHL). All studies were screened by two independent reviewers via the title/abstract/full text. The studies included any form of meditation/mindfulness/yoga as an intervention for RA. Animal studies, case reports, non-English articles, qualitative studies, conference abstracts, and articles without full-text access were excluded. Each study was assessed for its quality. Results: Out of 1527 potentially eligible records, 23 studies were included. All three MBTs showed various effects on patient-reported outcomes, such as vitality, functioning, and mental health, as well as on disease activity markers. Mindfulness-based interventions mainly reduced the subjective disease activity parameters (e.g., joint tenderness, morning stiffness, and pain), rather than the objective disease activity parameters (e.g., swollen joints and C-reactive protein (CRP)). RA patients with recurrent depression may benefit more from these non-pharmacological therapies than patients without recurrent depression. Discussion: This systematic literature review found that MBTs show added value in RA management, especially for patients with depressive symptoms. These non-pharmacological approaches, when used in addition to medication, might diminish polypharmacy in specific RA patient populations. **Lay Summary:** In recent decades, more attention has been given to the management of rheumatoid arthritis (RA) with options other than solely using medication. Such alternative options for patients to increase their quality of life are, for instance, meditation, yoga, and mindfulness. These examples of mind–body therapies (MBTs) are techniques that create an interaction between the mind and the bodily functions in order to obtain relaxation and enhance overall health. Although it is believed that these mind–body techniques are valuable in the management of RA, the extent of their contribution is still unclear, as is the question of if certain subgroups of patients benefit more from these complementary therapies. This systematic literature review investigated the effects of meditation, mindfulness, and yoga in patients with rheumatoid arthritis. A literature search was systematically performed within four different scientific databases by two independent reviewers. Out of 1527 potentially eligible articles, 23 studies were included. All three MBTs showed beneficial effects, which were mostly on the vitality, functioning, and mental health of patients with RA, but also on symptoms related to disease activity. RA patients with recurrent depression seemed to benefit more from these non-pharmacological therapies than patients without recurrent depression. Hence, we can conclude that MBTs show added value in the management of RA.

## 1. Introduction

Rheumatoid arthritis (RA) is a systemic autoimmune disease that typically causes a chronic joint inflammation. If left uncontrolled, this can lead to progressive joint damage, resulting in debilitating joint deformities [1]. The disease is also typically accompanied by manifestations such as pain, fatigue, and depression [2]. According to current guidelines, patients with early RA need to start treatment rapidly, and the therapeutic approach needs to be sufficiently intensive and adapted if needed according to a predefined disease activity target to reach the best possible outcomes [3]. Even though these strategies have been shown to be successful, a significant number of patients still experience an important impact of the disease on their physical, emotional, and social wellbeing. Therefore, non-pharmacological therapies can have an added value in the treatment of RA.

Over the last decades, there has been a substantial increase in studies regarding non-pharmacological management of RA, in addition to the pharmacological treatment thereof. These non-pharmacological approaches mainly include patient education, physical activity, relaxation, and meditation [4,5,6]. Meditation, yoga, and mindfulness are examples of mind–body therapies (MBTs), which are techniques that create an interaction between the mind and bodily functions to obtain relaxation and enhance overall health. All three of these MBTs have been shown to have multiple health benefits [7,8,9,10].

Although it is known that different forms of mind–body therapy (MBT) can be valuable in the management of RA, the extent of these contributions is still unclear, as is the identification of patients who benefit from these complementary therapies. Therefore, the aim of this systematic literature review (SLR) was to investigate the effects of meditation, mindfulness, and yoga in patients with RA.

## 2. Methods

### 2.1. Search Strategy

The search used in this SLR was systematically performed within 4 scientific databases (PubMed, Embase, Web of Science (core collection, Chinese and Korean collection), and CINAHL). Studies up to 18 July 2021 were included. Biomedical reference librarians of the KU Leuven Libraries—2Bergen—Learning Centre Désiré Collen (Leuven, Belgium) helped to create the search strings for each database, and they can be found in Appendix A. Each search string for each database was a combination of terms describing meditation, mindfulness, or yoga and RA. Endnote 20.1 was used to remove duplicates. Two independent reviewers (L.S. and D.D.C.) screened all studies with Rayyan QRCI, firstly based on the title/abstract and secondly based on the full text. If no consensus was reached, a third senior reviewer (K.D.M.) could be consulted. We checked for additional studies by snowballing in the references of the included full texts. The protocol for this SLR was registered on PROSPERO (reference number CRD42021264779).

### 2.2. Selection of the Included Studies

In the first step, a study was selected if the title/abstract included a term that potentially covered RA (e.g., arthritis or chronic inflammatory disease) in combination with an intervention that involved meditation, mindfulness, and/or yoga. There was no limit on the number of patients, the duration of the disease, or the length of the follow-up period. Neither animal studies nor case reports were considered. In addition, non-English articles and conference abstracts were excluded.

After screening based on the title/abstract, obtaining the full text was approached by using the extensive electronic collections of the KU Leuven, library requests, and personal communications with authors.

In the second step, the reviewers screened the studies based on the full texts. Qualitative studies and articles without full-text access were excluded. The final set of studies included forms of meditation, mindfulness, and/or yoga as interventions for patients with rheumatoid arthritis. The full-text screening excluded studies in which no distinction was made between patients with RA and other rheumatologic or chronic diseases.

### 2.3. Quality Assessment

The quality of the studies was assessed by means of three different tools.

The Methodological Index for Non-Randomized Studies (MINORS tool) was used for longitudinal studies, with maximal total scores of 16 and 24 for non-comparative studies and comparative studies, respectively [11]. There was one cross-sectional study, for which the Joanna Briggs Institute Critical Appraisal for Analytical Cross-Sectional Studies (JBI) was used, with a maximal global score of 8 [12]. The Physiotherapy Evidence Database (PEDro) was applied to randomized trials, and this tool was scored on a scale of 11 [13]. Higher scores on these quality appraisal tools indicated a higher study quality. A more detailed description of the different quality assessment tools can be found in Appendix A. The scores of each tool for each individual question in each study can be found in Appendix A.

## 3. Results

### Search Results

We obtained 585, 280, 481, and 181 references on PubMed, Embase, Web of Science (core collection, Chinese and Korean collection), and CINAHL respectively, resulting in a total of 1527 records. After removing the duplicates, 941 unique records remained. Screening based on the title/abstract resulted in a further reduction to 71 articles. After screening the full texts, 23 studies were finally included. No additional studies were found through snowballing. The flowchart of this systematic literature search is presented in Figure 1. We summarized each study and its main characteristics in Table 1. In total, 13 of the 23 (57%) studies were randomized controlled trials. Table 2 gives an overview of the different outcomes and tools for patient-reported outcomes used in each study. Table 2 divides the tools by outcome, including quality of life, functionality, fatigue, depression, anxiety, pain, stress, and sleep. Appendix A gives each study’s main conclusion(s) and most important limitation(s).

In this review, 14/23 (61%) of the included studies applied yoga as an intervention. According to most studies, yoga interventions seemed to improve the quality of life (measured by SF-36, WHOQOL, or EQ-5D) of patients with RA, as evidenced by better the outcomes in vitality, general health, physical functioning, and self-efficacy [16,17,18,20,21,22,23,24,25]. One study saw a better physical-health-related QoL and vitality when measured with Short Form-36 (SF-36), but found no effect on physical functioning as measured with the Health Assessment Questionnaire (HAQ) [20]. In contrast, another study found an effect on HAQ, but not on SF-36 [16].

Yoga has an as-yet-unclear effect on pain in patients with RA, as they showed improvements in pain scores, pain disability, pain intensity, acceptance of chronic pain, and self-efficacy regarding pain in four studies [17,18,19,21], although three studies found no change in pain levels [9,20,25]. Although multiple pain questionnaires were used (visual analog scales, the Simple Descriptive Pain Intensity Scale, the Pain Disability Index, and the Wellbeing Measurement Framework survey), no consistency could be traced. Similarly, two studies showed improved fatigue scores [21,25], while two other studies did not find changes in fatigue scores [9,22].

Out of the studies that analyzed the effects of yoga on mental health, 67% (4/6) observed that symptoms of depression in RA patients decreased significantly after yoga interventions [17,18,21,25], while in two studies, this effect was not found [9,22]. Puksic et al. (2021) described an improvement in anxiety levels according to the Hospital Anxiety Depression Scale [25], but according to Evans et al. (2010), the decrease in anxiety was not present when assessed with the Wellbeing Measurement Framework survey [18].

Yoga interventions caused a decrease in RA disease activity markers, with most of the studies showing an improvement in DAS28 scores, erythrocyte sedimentation rate (ESR), number of inflamed joints, morning stiffness, C-reactive protein (CRP), rheumatoid factor (RF), lymphocyte count, and serum uric acid [16,19,20,24]. This is in line with the improvement in hand grip strength for RA patients practicing yoga [14,15,20]. However, two studies noted no improvement in DAS28 scores [21,25]. A downregulation of pro-inflammatory markers (IL-1a, IL-6, TNF-α, and CTLA4) and an upregulation of anti-inflammatory markers (TGF-β) were observed [10,23]. In addition to the reduction in inflammatory cytokines, yoga elevates mind–body communicative markers (BDNF, DHEAS, β-endorphin, and sirtuins), indicating an effect on the psycho-neuro-immune axis [23].

Two studies found a positive effect on heart rate and systolic and diastolic blood pressure in patients with RA after practicing yoga [10,19], while one study found no changes in heart rate [17]. Yoga improved mitochondrial health according to the better cortisol, melatonin, and serotonin levels [10,24]. In contrast, one study did not find an effect on cortisol levels [17].

Six of the 23 (26%) included studies used mindfulness-based interventions (MBIs). These are mindful awareness and acceptance treatment [27], mindfulness-based cognitive therapy (MBCT) [29], mindfulness-based emotion regulation therapy [26], and mindfulness-based stress reduction (MBSR) [6,28,30].

Mindfulness interventions seemed to influence pain perception in patients with RA, with improved pain scores as measured with the visual analog scale, but also by proxy outcomes, such as pain-related catastrophizing and pain coping efficacy, as seen in the studies exploring the impact of mindfulness on pain [26,27,28]. This positive effect on pain was in line with the decrease in duration of morning stiffness and joint tenderness established by mindfulness interventions [27,28]. However, mindfulness did not cause an improvement in RA disease activity markers, such as CRP and swollen joint count [6,28].

Mindfulness helped to reduce stress [6,27,29]. This decrease in psychological distress can be extended to the positive effect of mindfulness on wellbeing, depression, and anxiety [6,27,29]. Two studies found that the emotion-regulating aspects of mindfulness were more beneficial to RA patients with recurrent depression than to those without it. This was indicated by a greater improvement in several outcomes, such as changes in pain-related fatigue and joint tenderness, on one hand, and changes in stress-related positive and negative affect, as measured by the Positive and Negative Affect Schedule (PANAS), on the other hand [26,27]. Another study found that the effect of MBSR on RA disease activity may be mediated in a positive manner by improvements in depressive symptoms [30]. Dalili et al. (2019) found that mindfulness-based interventions helped to improve illness perception scores in people with RA [29].

Three out of the 23 (13%) studies examined the effect of meditation, and they reported on progressive muscle relaxation (PMR) or the Benson Relaxation Technique (BRT) [31,32,33].

PMR seemed to improve physical function, vitality, social function, mental health, and general and psychological health, but not bodily pain or physical constraint [33]. In contrast, Barsky et al. (2010) integrated PMR into a relaxation response training. Pain was improved, but other symptoms, such as stiffness, swelling, restriction of movement, fatigue, and malaise, were not alleviated [32]. Neither depression nor anxiety were improved [32].

The BRT seemed to have a positive effect on wellbeing, anxiety, morning stiffness duration, ESR, and depression in patients with RA. The effects on hemoglobin and platelet levels, fatigue, RF, CRP, and the indexes for restrictive, painful, and swollen joints were not significant in patients with RA after the BRT [31].

## 4. Discussion

In this SLR, we studied the effects of meditation, mindfulness, and yoga on patients with RA. We found that all three MBTs have various effects on multiple domains—not only on patient-reported psychosocial outcomes, but also on disease activity markers and scores. RA patients with recurrent depression may benefit more from mindfulness techniques than patients without recurrent depression [26,27].

We found that mind–body interventions improve both biological and psychosocial outcomes in patients with RA, such as vitality, functioning, and general health, resulting in a better quality of life [16,17,18,20,21,22,23,24,25,31,33]. This result is in agreement with previous findings regarding other auto-immune diseases, such as inflammatory bowel disease (IBD) [34,35], but healthy adults also benefit from mind–body interventions [36].

Yoga—more than mindfulness and meditation—seems to reach better outcomes in terms of RA disease activity markers, such as DAS28 scores, morning stiffness, affected joints, CRP, and inflammatory markers [10,16,19,23,24]. On the other hand, mindfulness interventions seem to improve pain perception, the duration of morning stiffness, and joint tenderness [26,27,28], but do not seem to establish improvements in objective RA disease activity parameters (e.g., CRP and swollen joint count) [6,28]. The effects of yoga interventions on pain are less obvious. Four studies found a positive effect on pain-related outcomes [17,18,19,21], but three studies found no changes [9,20,25].

MBTs are also beneficial for the mental health of RA patients, as shown by the improvement in depressive symptoms and anxiety [6,17,18,21,25,27,29,31,33]. Two studies found that practicing mindfulness also had greater effects on RA patients with recurrent depression [26,27]. Again, this effect of MBTs on mental health was also seen in adults with IBD, as shown by a reduction in depression and anxiety scores [34].

Fogarty et al. found that improvements in depression may positively mediate the effect of mindfulness on RA disease activity, suggesting that the beneficial effect of mindfulness on RA may be explained by a decrease in depressive symptoms [30]. Changes in RA disease activity are temporarily influenced by depressive symptoms, with slower improvements in global assessments, tender joints, and patient-reported pain and function [37]. Further research should focus on the mechanism of interaction between mindfulness-related reductions in depressive symptoms and improvements in RA symptoms to optimize the effects of MBTs in this patient group.

Overall, the heterogeneity in types of interventions, outcomes and outcome tools across the 23 studies included in this SLR was remarkable.

Firstly, the studies did not always differentiate between the three MBTs. For each intervention (i.e., meditation, mindfulness, and yoga) there were, again, various subtypes. For instance, mindfulness was carried out in the form of mindful awareness and acceptance treatment [27], MBCT [29], mindfulness-based emotion regulation therapy [26], and MBSR [6,28,30]. Some studies combined different MBTs into one other MBT; for example, the study of Singh et al. (2011) integrated meditation into their ‘yogic package’ intervention [19].

Secondly, there were many different outcomes and outcome tools across and even within studies. This made it challenging to compare the results of all reports. For example, 8 different tools in 11 studies examined changes in depression (see Table 2).

Some limitations challenge the robustness of the evidence in this SLR. Firstly, most studies had a modest sample size, with N ≤ 51 in 11 of them. Secondly, there was a wide variability in the duration of the interventions, the (healthcare) professionals who applied the interventions, and the follow-up periods. The long-term effects of these interventions were mostly not researched, though the long-term effects of MBTs could be interesting. Longer mindfulness practice is, for example, associated with structural effects on brain areas related to pain perception [38].

Most of the studies were carried out in the USA (8/23) and India (6/23), but there are differences in compliance with and acceptability of mind–body interventions across different cultures and healthcare settings. The patients included in the studies were also willing to participate in these mostly unblinded mind–body interventions; thus, they had greater intrinsic motivation, which could lead to bias.

Moreover, the included studies were executed between 1994 and now, with only eight being relatively recent. The added value of mind–body interventions might differ over time as treatment strategies improve. Finally, the studies that were included sometimes differed in their results, and even results in the same study could be found to have opposite meanings. Interpretations of summaries of the results, as we tried to give in this systematic review, are, thus, challenging. Each study should, therefore, be carefully examined. In Appendix A, we included each study’s main conclusion and most important limitations in order to aid the reader.

In conclusion, this SLR confirms that non-pharmacological treatment methods have an added value in the management of RA. Based on the current evidence, MBTs seem valuable for many RA patients, especially for those with depression. This SLR, which considered the effects of meditation, mindfulness, and yoga on RA, revealed considerable heterogeneity across and within studies with respect to sample size, outcome tools, and interventions. Thus, this research field needs studies with larger sample sizes, more uniform outcome tools, and standardized interventions. In future management of RA, we recommend paying more attention to these non-pharmacological approaches. This can be one of the ways to diminish polypharmacy in RA patients, especially in the vulnerable patient subpopulation with psychological distress, with the aim of minimizing adverse events and costs for society.

## Figures and Tables

**Figure 1 jpm-12-01905-f001:**
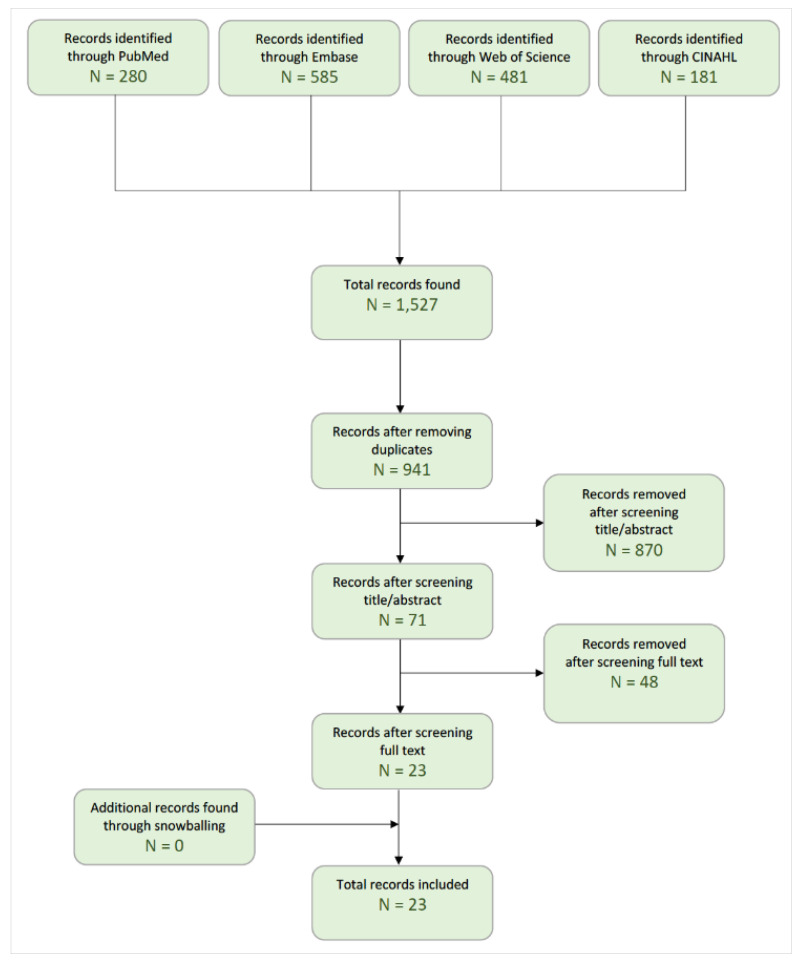
Flowchart of the systematic literature review process.

**Table 1 jpm-12-01905-t001:** Characteristics of the included studies, interventions, and participating patient populations.

Author (Year)	Intervention	Country	Sample	Mean/Median Age	Disease Duration	Women (%)
**Yoga**						
Haslock (1994) [14]	Yoga	UK	N = 20	55 years	17 years	85%
Dash (2001) [15]	Yoga	India	N = 40	33 years	NA	50%
Badsha (2009) [16]	Yoga	UAE	N = 47	45 years	73 months	NA
Bosch. (2009) [17]	Yoga	USA	N = 16	61 years	19 years	100%
Evans (2010) [18]	Iyengar Yoga	USA	N = 5	28 years	NA	80%
Singh (2011) [19]	Yogic package (incl. meditation)	India	N = 80	35 years	12 months	70%
Telles. (2011) [20]	Yoga	India	N = 64	47 years	NA	73%
Evans. (2013) [21]	Iyengar Yoga	USA	N = 30	28	NA	80%
Ward (2018) [9]	Yoga	New Zealand	N = 26	54y	12	96%
Greysen (2019) [22]	Yoga	USA	N = 398	62y	25	88%
Gautam (2020) [23]	Yoga-based lifestyle	India	N = 66	44	NA	80%
Ganesan (2020) [10]	Yoga	India	N = 166	42	NA	92%
Gautam (2021) [24]	Yoga	India	N = 70	45	6	80%
Puksic (2021) [25]	Yoga	Croatia	N = 57	55	8	95%
**Mindfulness**						
Pradhan (2007) [6]	MBSR	USA	N = 63	54	9	87%
Zautra (2008) [26]	MERT	USA	N = 144	54	13	68%
Davis (2015) [27]	MAAT	USA	N = 143	54	NA	69%
Fogarty (2015) [28]	MBSR	New Zealand	N = 42	54	11	88%
Dalili (2019) [29]	MBCT	Iran	N = 28	NA	NA	NA
Fogarty (2019) [30]	MBSR	New Zealand	N = 51	54	11	88%
**Meditation**						
Bagheri-Nesami (2006) [31]	Benson Relaxation	Iran	N = 50	49	6	96%
Barsky (2010) [32]	Relaxation Response (PMR)	USA	N = 168	53	13	87%
Yazdani (2017) [33]	PMR	Iran	N = 62	NA	NA	NA

N, number of patients; NA, not available; MBCT, mindfulness-based cognitive therapy; PMR, progressive muscle relaxation; MBSR, mindfulness-based stress reduction; MAAT, mindful awareness and acceptance treatment.

**Table 2 jpm-12-01905-t002:** Patient-reported psychosocial outcomes and tools in each study.

Author (Year)	QOL	ADL	Fatigue	Depression	Anxiety	Pain	Stress	Sleep
**Yoga**								
Haslock (1994) [14]		HAQ						
Dash (2001) [15]								
Badsha (2009) [16]	SF-36	HAQ						
Bosch. (2009) [17]		HAQ		BDI				
Evans (2010) [18]	SF-36	HAQ		BSI-18	BSI-18	PDI		
Singh (2011) [19]						SDPIS		
Telles. (2011) [20]		HAQ						
Evans. (2013) [21]	SF-36	HAQ	FACIT	WMF	WMF	PDI, WMF		WMF
Ward (2018) [9]	EQ-5D-3L	HAQ	BRAF-NRS	HADS.D	HADS.A	VAS		ISI
Greysen (2019) [22]		HAQ				VAS		PROMIS
Gautam (2020) [23]	WHOQOL							
Ganesan (2020) [10]								
Gautam (2021) [24]		HAQ						
Puksic (2021) [25]	SF-36		FACIT	HADS.D	HADS.A		PSS	
**Mindfulness**								
Pradhan (2007) [6]	Wellbeing scales			SCL-90-R				
Zautra (2008) [26]						VAS		
Davis (2015) [27]				PANAS-X	PANAS-X			
Fogarty (2015) [28]						VAS		
Dalili (2019) [29]	IPQ-R			DASS	DASS		DASS	
Fogarty (2019) [30]				HADS.D	HADS.A			
**Meditation**								
Bagheri-Nesami (2006) [31]				BDI	SSTAI			
Barsky (2010) [32]	AIMS			MHI	MHI	RASQ		RASQ
Yazdani (2017) [33]	SF-36							

SF-36, Medical Outcome Survey Short Form-36; FACIT, Functional Assessment of Chronic Illness Therapy; HADS, Hospital Anxiety Depression Scale for assessment of depression (HADS.D) and anxiety (HADS.A); PSS, Perceived Stress Scale; BDI, Beck Depression Inventory; PANAS-X, Positive and Negative Affect Scale—Expanded Form; BRAF-NRS, Bristol Rheumatoid Arthritis Fatigue Numerical Rating Scales; EQ-5D-3 L, EuroQoL; HAQ(-DI), Health Assessment Questionnaire (Disability Index); ISI, Insomnia Severity Index; VAS, Visual Analogue Scale; PROMIS, Patient-Reported Outcome Measurement Information System; IPQ-R, Revised Illness Perception Questionnaire; DASS, Depression Anxiety Stress Scales; RASQ, Rheumatoid Arthritis Symptom Questionnaire; MHI, Mental Health Inventory; AIMS, Arthritis Impact Measurement Scale; SDPIS, Simple Descriptive Pain Intensity Scale; PDI, Pain Disability Index; BSI-18, Brief Symptom Inventory; WHOQOL-BREF, WHO Quality of Life BREF; WMF, Wellbeing Measurement Framework survey; SCL-90-R, The Symptom Checklist-90-Revised; SSTAI, Spielberger’s state–trait anxiety index.

## Data Availability

Not applicable.

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
