# Peer review of "The Effect of Meditation, Mindfulness, and Yoga in Patients with Rheumatoid Arthritis"

_jpm, 2022, doi:10.3390/jpm12111905_

Round 1

Reviewer 1 Report

Thanks for the opportunity to review the article: The effect of meditation, mindfulness, and yoga in patients with rheumatoid arthritis

I have some comments on this interesting paper.

Introduction

No comments

Material and methods

I think that the authors could add the search terms that were used to carry out the systematic search.

The authors describe the following: "The Methodological Index for Non-Randomized Studies (MINORS tool) was used for longitudinal studies, with a maximal total score of 16 and 24,". However, it would be interesting if the authors described the cut-off point for the choice of investigations.

Results

Although a short description of the studies included in the study is made. I believe that the quality of the work could be improved if the authors added a table with the main results of the studies included in the review. For example, the table could summarize the following: "We found that mind-body interventions improve both biological and psychosocial outcomes in patients with RA, such as vitality, functioning and general health, resulting in a better quality of life." This suggestion applies to the rest of the results.

Discussion

No comments

Author Response

We would like to thank reviewer 1 for his positive review of our systematic review on meditation, mindfulness, and yoga in patients with rheumatoid arthritis

  1. I think that the authors could add the search terms that were used to carry out the systematic search.

We agree with reviewer 1 that giving the algorithms used to search the different databases is a prerequisite for every published systematic review. This document giving a detailed view of the search string per database can be found in supplementary file (part A). We also included in the methods section the following sentence:

“Biomedical reference librarians of the KU Leuven Libraries – 2Bergen – learning Centre Désiré Collen (Leuven, Belgium) helped to create the search strings per database, which can be found in supplementary file A. Each search string per database was a combination of terms describing either meditation, mindfulness, or yoga, and RA.”

  1. The authors describe the following: "The Methodological Index for Non-Randomized Studies (MINORS tool) was used for longitudinal studies, with a maximal total score of 16 and 24,". However, it would be interesting if the authors described the cut-off point for the choice of investigations.

We thank reviewer 1 for this interesting suggestion. It is challenging to give a clear threshold for a study with a good quality. The MINORS tool is a summary of 12 different concepts: 1. clearly stated aim; 2. inclusion of consecutive patients; 3. prospective data collection; 4. endpoints appropriate to study aim; 5. unbiased assessment of study endpoint; 7. <5% lost to follow-up; 8. prospective calculation of study size; Additional criteria in the case of comparative study: 9. adequate control group; 10. contemporary groups; 11. baseline equivalence; 12. adequate statistical analyses. For the included studies, we could clearly see that most information was missing for the unbiased assessment of study endpoint and the prospective calculation of study size.

For readers interest in this we included for all quality assessment tools and analysis all the details in supplementary files B & C.

“A more detailed description of the different quality assessment tools can be found in supplementary file B. Scores per study per individual question per tool can be found in supplementary file C.”

  1. Although a short description of the studies included in the study is made. I believe that the quality of the work could be improved if the authors added a table with the main results of the studies included in the review. For example, the table could summarize the following: "We found that mind-body interventions improve both biological and psychosocial outcomes in patients with RA, such as vitality, functioning and general health, resulting in a better quality of life." This suggestion applies to the rest of the results.

This is indeed a great suggestion by reviewer 1 to improve the interpretation of our results. We have made a supplementary file D which includes details of the main conclusion per included article and also main limitations as suggested by reviewer 2.

The text was adapted in the following way:

“Table 2 gives an overview of the different outcome tools for patient-reported outcomes used in each study. Table 2 divides the tools per outcome including quality of life, functionality, fatigue, depression, anxiety, pain, stress, and sleep. Supplementary file D gives per study its main conclusion and most important limitations.”

Reviewer 2 Report

The  authors ' aim is to verify the effect of meditation, mindfulness and yoga in patients with rheumatoid arthritis with a statistical approch on criteria of screnning applied in litterature data derived from PuBMed, Embase Web Scince CINAHL. In the end authors  utilize for their work ONLY a total of 23 articles. 14 on 23 references applies yoga as intervention, six on 23 include studies which  used mindfulness as intervention and tre on 23 studies examine the effect of meditation.

English is fine and althoght the criteria and statistical approch  in general it  seems good, pheraps this is the sole purpose of authors,  I belive that the contribution of this article,  only from the scientific point of view is low for the reasons below elencated 

1) About the few articles selected, many are old and only 8 are more recent, moreover in some of them conclutions  in contrast are reported . About scientific conclusion and discussion, a more caution would be apportune and I invite the authors modifies these sections , taking into acount and mentioning these issues.

2) Moreover  in some articles included in this work,  limitations in approch are elencated  . I invite the authors to expose them

I accept the work after these modifications. 

Author Response

Reviewer 2:

We would like to thank reviewer 2 for his constructive review of our systematic review on meditation, mindfulness, and yoga in patients with rheumatoid arthritis

  1. The  authors ' aim is to verify the effect of meditation, mindfulness and yoga in patients with rheumatoid arthritis with a statistical approch on criteria of screnning applied in litterature data derived from PuBMed, Embase Web Scince CINAHL. In the end authors  utilize for their work ONLY a total of 23 articles. 14 on 23 references applies yoga as intervention, six on 23 include studies which  used mindfulness as intervention and tre on 23 studies examine the effect of meditation.

We thank reviewer 2 for his assessment of our systematic literature review. In total, 941 articles could be screened. It shows that research in this field is still quite limited, as most systematic reviews we undertake result in more hits. This also explains the limited size of included articles. However, as we worked together with experienced medical librarians, this gives us confident that we could include all relevant works for this review.

  1. English is fine and althoght the criteria and statistical approch  in general it  seems good, pheraps this is the sole purpose of authors,  I belive that the contribution of this article,  only from the scientific point of view is low for the reasons below elencated.

We want to thank reviewer 2 for his kind words regarding text and approach. We also agree that perhaps the overarching conclusions are limited from a practical point of view as the heterogeneity in design and outcome per included study is great. Therefore, we also refrained from doing a meta-analysis.

  1. About the few articles selected, many are old and only 8 are more recent, moreover in some of them conclutions  in contrast are reported . About scientific conclusion and discussion, a more caution would be apportune and I invite the authors modifies these sections , taking into acount and mentioning these issues.

Indeed, reviewer 2 is correct that bias in a such a way can exist. Therefore we have added to our limitation section in the discussion the following paragraph:

Moreover, the included studies were executed between 1994 and now, with only 8 being relatively recent. The added value of mind-body interventions might differ over time as treatment strategies have improved. Finally, the studies included sometimes differed in result, and even results in the same study could be found to have opposite meanings. Interpretation of a summary of the results as we tried to give in this systematic review are thus challenging. Each study should therefore be carefully examined. We included in Supplementary file D per study its main conclusion(s) and most important limitation(s) to aid the reader.

  1. Moreover  in some articles included in this work,  limitations in approch are elencated  . I invite the authors to expose them.

We have tried to accommodate the comment of reviewer 2 and added a supplementary file D that gives details per study on conclusion and outcome.

The text was adapted in the following way:

“Table 2 gives an overview of the different outcome tools for patient-reported outcomes used in each study. Table 2 divides the tools per outcome including quality of life, functionality, fatigue, depression, anxiety, pain, stress, and sleep. Supplementary file D gives per study its main conclusion and most important limitations.”

Round 2

Reviewer 1 Report

Thanks again for the opportunity to review the manuscript: The effect of meditation, mindfulness, and yoga in patients with rheumatoid arthritis: "The effect of meditation, mindfulness, and yoga in patients with rheumatoid arthritis"

The authors have responded to each of my comments satisfactorily. I consider that the article can be accepted for publicatio